# Macrophyte-Based Assessment of Upland Rivers: Bioindicators and Biomonitors

**DOI:** 10.3390/plants12061366

**Published:** 2023-03-19

**Authors:** Gana Gecheva, Silviya Stankova, Evelina Varbanova, Lidia Kaynarova, Deyana Georgieva, Violeta Stefanova

**Affiliations:** 1Faculty of Biology, Plovdiv University, 4000 Plovdiv, Bulgaria; 2Faculty of Chemistry, Plovdiv University, 4000 Plovdiv, Bulgaria

**Keywords:** aquatic macrophytes, bryophytes, mosses, ecological status, EQS

## Abstract

For the first time, a macrophyte-based assessment of ecological status was related to the accumulated heavy metals and trace elements (Al, As, Cd, Co, Cr, Cu, Fe, Hg, Mn, Ni, Pb, Zn) in aquatic plants. Three moss and two vascular plant species were applied as biomonitors: *Fontinalis antipyretica* Hedw., *Leptodictyum riparium* (Hedw.) Warnst., *Platyhypnidium riparioides* (Hedw.) Dixon, invasive *Elodea canadensis* Michx., and *Myriophyllum spicatum* L. Three streams were assessed as good at a high ecological status which correlated with low contamination based on calculated contamination factors (CFs) and metal pollution index (MPI). Two sites evaluated in moderate ecological status were revealed to be in heavy trace element contamination. The most significant was the accumulation of moss samples from the Chepelarska River under mining impact. Mercury exceeded the environmental quality standard (EQS) for biota in three of the studied upland river sites.

## 1. Introduction

The main stressors regarding upland rivers are related to the growing need for water resources for irrigation, drinking water supply, and industry (with the processes of erosion, hydromorphological destruction of aquatic habitats, pollution related to untreated domestic wastewater, as well as due to mining activities). As a result, river degradation has adverse effects on aquatic flora and fauna.

Aquatic macrophytes (aquatic bryophytes and vascular plants) are a vital part of freshwater ecosystems. They provide an important physical substrate for periphyton, habitat, and refuges for benthic macroinvertebrates and fish [1]. Their role in the stream functional processes, such as nutrient cycling and metabolism, is extremely important.

Aquatic macrophytes also have a species-specific sensibility to environmental conditions, and thus the presence or absence of particular species can be used to classify river status. The history of aquatic macrophyte application to assess stream responses to anthropogenic stressors dates back to the 1960s. In parallel, the term bioindicator was also introduced into the literature. It can be considered as an organism (or part of an organism or a group of organisms) that contains information on the quality of the environment [2]. Freshwater quality can be reflected by individual species’ abundance and by the community structure. A recent study on upland rivers documented that stressors (hydropeaking, organic, and inorganic pollution) and drivers (discontinuous urban structures) modify aquatic macrophyte assemblages in terms of abundance and biological and ecomorphological types [3]. Current water policy, particularly the Water Framework Directive (WFD), [4] requires Member States to assess the ecological status of water bodies based on biological quality elements, including aquatic macrophytes. Ecological status is presented by Ecological Quality Ratio (EQR) with a range between 0 and 1, where 1 reflects the highest ecological quality. Most macrophyte-based assessment systems in Europe detect eutrophication as a main pressure [5]. Reference Index (RI) which reflected multiple stressors has been adopted and applied in Bulgaria [6]. Bryophytes represent a significant part of indicators in macrophyte-based RI for ecological status assessment, especially in upland rivers where they have a large biomass and high production level.

Parallel to bioindication, in the last century, significant attention was paid to biomonitoring and especially to bryophytes as monitors of heavy metal pollution [7]. The main reasons are high bioaccumulation capacity and reflection of the soluble metal fraction in the environment, which is likely to affect the major compartments of the aquatic ecosystem. The most commonly used and proven biomonitors in European freshwater ecosystems are *Fontinalis antipyretica*, *Platyhypnidium riparioides,* and, to a lesser extent, *Leptodictyum riparium* [7,8]. Aquatic vascular plants, particularly submerged *Myriophyllum spicatum* and invasive *Elodea canadensis*, are less frequently a subject of biomonitoring studies in aquatic environments. Nevertheless, macrophytes’ pollution tolerance and their application in phytoremediation have been studied [9]. Eurasian watermilfoil was recommended as an effective biosorbent for the removal of zinc, lead, and copper [10]. High resistance of *Elodea canadensis* and *Myriophyllum aquaticum* to heavy metals and sorption via surface directly from the water was reported [11].

Environmental Quality Standards Directive (EQSD) requires high-quality monitoring of information on the concentrations of polluting substances in the aquatic environment [12]. Environmental Quality Standards (EQSs) are thresholds defining a good chemical status and a guarantee that there is no risk to aquatic ecosystems and human health. Only the biota standard for mercury has been included among inorganic priority substances up until now.

Considering the dualistic role of aquatic plants as indicators and biomonitors, in 2021, we conducted a survey of six upland rivers in Bulgaria (southeastern Europe). In parallel, an assessment of the ecological status under RI was made and samples were taken for bioaccumulation. The purpose was to illustrate the complementary nature of aquatic macrophyte application both as a part of ecological status metrics and as a matrix for priority substance monitoring. If our hypothesis is correct, and if upland rivers (where bryophytes are the dominant flora) are under toxic stressors, then accumulation in biota should be added to the final assessment of the ecological status. The hypothesis implicit is that, in the case of chemical pollution to which mosses are tolerant, the macrophyte-based index should be supported with accumulation in biota data.

## 2. Results

Studied upland rivers were small, up to 5 m wide, with a large number of stones and rapid currents. River waters were slightly alkaline, with predominantly low conductivity (Table 1). Measured physicochemical parameters (conductivity and dissolved oxygen) corresponded to high status, except for the first river site.

### 2.1. Aquatic Macrophytes as Bioindicators

Twenty-eight taxa from two taxonomic groups (bryophytes, spermatophytes) were registered (Table 2). The Hygrophyte group was the most diverse group and was represented by 12 species. The total number of species per site varied between 2 and 10 taxa (median 6). Macrophyte abundance was relatively low, which is typical for upland streams, with a median of 28. The richest in taxa and abundance macrophyte community was described by the true hydrophytes *Ceratophyllum demersum*, *Elodea canadensis*, *Myriophyllum spicatum*, *Ranunculus aquatilis* (Table 3). The most common emergent species were helophytes *Berula erecta*, and *Typha latifolia*.

From the recorded taxa, thirteen macrophytes are listed in three indicator groups according to the adapted Reference Index [6], including six aquatic mosses.

*Platyhypnidium riparioides*, an indicator of undisturbed habitats (Group A), dominated communities at two sites (3 and 4; Table 3) with the highest Ecological Quality Ratio (EQR). Vascular plants from the disturbance indicators’ group C (*E. canadensis*, *M. spicatum*, *Potamogeton crispus*) dominated at sites assessed in moderate status (sites 1 and 6, Table 3). These two semi-mountain river sites were significantly affected by abstraction. The first site was also characterized by destroyed riparian vegetation and high toxic risks. At this site, the aquatic moss *Leptodictyum riparium* from the group of independent taxa (Group B) was also recorded as a dominant species.

### 2.2. Aquatic Macrophytes as Biomonitors

Three moss and two vascular plant species were applied as biomonitors: *Fontinalis antipyretica* (site 5), *Leptodictyum riparium* (sites 1 and 2), *Platyhypnidium riparoides* (sites 3 and 4), *Elodea canadensis*, and *Myriophyllum spicatum* (site 6). Both *E. canadensis* and *M. spicatum* are submerged rooted perennial plants, with reproduction mainly by vegetative fragmentation.

The inorganic composition of the plant samples (Table 4) illustrated the strongest element variation for Cd (323 times), followed by Pb (270) and Zn (58). Copper bioaccumulation varied between samples 15 times (Cr—11, Mn and Ni—8, Al and As—5, Fe—4, and Co—3 times). Mercury concentrations in plant samples were below the Methodological Limit of Determination (0.05 mg kg^−1^) at four sites (sites 2,3, 4, and 6), while the two moss samples at sites 1 and 5, and *M. spicatum* at site 6 exceeded the environmental quality standard (EQS) for biota (0.02 mg kg^−1^).

The obtained percent recoveries (R%) with corresponding confidential intervals for the reference materials were in the following ranges: 106 ± 8–111 ± 7% for Al; 87 ±–110 ± 16% for As; 97 ± 12–105 ± 5% for Cd; 99 ± 10–112 ± 7% for Co; 93 ± 4–113 ± 9% for Cr; 93 ± 4–100 ± 8% for Cu; 92 ± 6–107 ± 7% for Fe; 99 ± 5–102 ± 6% for Mn; 101 ± 9–104 ± 11% for Ni; 95 ± 4–102 ± 6% for Pb; and 95 ± 6–100 ± 4% for Zn. Considering both—the large variation in the concentrations of the investigated elements as well as the different matrix composition of the used reference materials, the obtained recoveries proved that the developed analytical methods are capable to provide true and adequate information about the elemental composition of the tested real samples. Among 5 applied biomonitors, *Leptodictyum riparium* (sampled from site 1) accumulated the highest amount of all 12 studied elements.

The two vascular plant species, *E. canadensis* and *M. spicatum*, sampled at site 6, revealed no significant differences in the bioaccumulation rates.

Among the 12 studied elements, Al had the highest accumulation levels in mosses, followed by Fe, Mn, Zn, and Cu, except for the sample from site 3, where in *Platyhypnidium riparoides* tissues manganese led the decreasing element order. Iron was the dominant element in the two vascular plant samples. Cadmium in both moss and vascular plant samples was the element with the lowest accumulation levels, except for site 1 under mine and industrial impact.

Based on the field observation (no abstraction and channelization, no direct alteration of the riparian vegetation and in-stream habitats, no toxic risk), measured basic physicochemical parameters, and assessed ecological status, site 3 was selected as a near reference. Analyzed elements in the moss sample from the site were assumed as background levels. Heavy metal and toxic element concentrations (Al, As, Cd, Co, Cr, Cu, Fe, Mn, Ni, Pb, and Zn) were converted to CFs and MPI (Table 5).

Major contaminants appeared to be Pb and Cd (highest CFs), as well as Zn, Cu, and Cr, resulting in extreme pollution along Chepelarska and Stryama rivers, and moderate pollution at Shirokolashka River. Two of the studied upland river sites appeared to be of low contamination.

## 3. Discussion

### 3.1. Aquatic Macrophytes as Bioindicators

Aquatic macrophyte communities dominated by mosses (*P. riparioides* and *F. antipyretica*) forming large carpets, represented undisturbed assemblages in upland rivers as previously reported [3]. As found in the cited study, these typical upland reference macrophyte assemblages are displaced by richer taxa and more abundant communities of helo- and hydrophyte species character for lowland rivers such as *C. demersum* and *M. spicatum*. Species composition and the indicators’ abundance at river sites assessed in moderate status suggested that these habitats are affected. Additionally, species uncharacteristic of upland rivers occurred (e.g., *Typha latifolia*). The record of *Leptodictyum riparium* at these sites confirmed species tolerance to disturbance and its application as a signal species, indicating hydromorphological pressure and pollution [3]. Our records of the species once again confirmed its relation to the indifferent indicators’ group B. The moss has a wide ecological niche and attaches to a wide spectrum of substrates: underwater stones, wood, and organic matter. It was also found on moist or wet soil along the shore.

Canadian waterweed has high ecological tolerance and is found in waters with average nutrient content, ranging from mesotrophic to eutrophic status [14]. During the current study, it was recorded together with other vascular macrophytes in an unshaded semi-mountain site assessed at a moderate status. This was in confirmation with found species preferences for sunny river sections located up to 500 m a.s.l. and classified mainly in good and moderate statuses in Poland.

### 3.2. Aquatic Macrophytes as Biomonitors

Applied as biomonitors, native rheophilic moss species, as well as submerged vascular plants, have a proven high sensibility to different stressors and a wide distribution [7,15].

Trace elements’ background concentrations in *P. riparioides* at reference site 3 were similar or lower compared to concentrations in the same species from another reference site in Bulgaria [16], while Al, Fe, Mn, and Zn were three to four times higher.

Comparison with data for the same species sampled from the mine region in Central Portugal [17] showed that in *P. riparioides* from the Rhodopes, Bulgaria the elements have lower values, from seven times for Ni to two times for Zn and Pb. The only exception is Mn, having 23 times higher levels in studied streams in Rhodopes (7837.8 mg kg^−1^). Similar enrichment with Mn detected with terrestrial mosses was reported from West Serbia as a result of heavy industry and intense traffic [18]. Manganese is an important microelement and generally beneficial to plants, and less toxic. It was found to be highly variable in aquatic bryophytes—between 35,650 mg/kg and 77 mg/kg [19]. The same study reported the capacity of aquatic mosses to accumulate significant amounts of Fe, Al, and Mn in their tissues, which was confirmed by our results.

Comparison between *F. antipyretica* at site 5, an unimpacted area in Rila Mountain sampled within the current study and with previous research in 2001 [19], revealed that for the period of 20 years, the elements Co, Cu, Ni, and Pb diminished about 3 times, while Al, Fe, and Mn maintained similar levels.

The documented extreme aquatic moss pollution at Chepelarska River (before inflow into Maritsa River, east of Plovdiv) confirmed reports of previously high levels of heavy metal pollutants in the river’s water [20] and the fact that it is still under mining impact. A similar accumulation of Cd (63 mg kg^−1^) and Pb (1054 mg kg^−1^) in mosses was reported from Topolnitsa River Basin (Bulgaria) under copper production industry impact [16]. *L. riparium* sampled 20 years ago from Maritsa River, after Chepelarska River inflow, had significantly lower values for As (7 times), Pb (4 times), and Al (3 times) [19]. The dilution of pollutants at the inflow into the larger river and the time for self-purification downstream should be taken into account.

As for the aquatic vascular plants, tissue levels of toxic metals Cd, Pb, and Hg were found to be at least one order of magnitude greater than ambient water and sediments [21]. The highest degree of accumulation was registered for Mn, Zn, Cu, and Pb. Samples from *E. canadensis* at the second river site in extreme pollution Stryama River (Table 4) contained elevated levels of both priority substances, Cd and Pb. The stream could be affected by the waste dump (unregulated industrial waste) near Karlovo town, which is located in close proximity to the Stara River, a tributary of the Stryama River. Similar Cd accumulation in *E. canadensis* was reported in a contaminated metal industry pond in north-eastern France [22], but the analyzed lead concentration is nearly three times lower compared to our results (10.1 mg kg^−1^). The potential of *E. canadensis* for river pollution monitoring, especially for Al, Cr, and Cu, was reported [15]. Based on the results obtained, we could also state a proven species potential for the toxic Pb and Cd.

Comparison between the two aquatic vascular plant species applied as biomonitors showed no deference in element content between *E. canadensis* and *M. spicatum*. The determined mercury of 0.052 mg kg^−1^ in *M. spicatum* is of interest, while it was below the MLD in *E. canadensis* (<0.05 mg kg^−1^). Even though most of the plants accumulate mercury in their roots [23], our results for the upper parts of the basal shoots showed that *M. spicatum* is able to accumulate amounts of Hg in their shoots too, and this supports the findings for another *Myriophyllum* species [11].

Mercury EQS for biota excess was registered in samples from the above two impacted sites, as well as in site 5 located in the relatively pristine region. The most abundant species applied as biomonitor at the last site was *Fontinalis antipyretica*. The sampling location was in an undisturbed area, where atmospheric deposition could be suggested as a source of contamination. A similar median Hg concentration (101 ng g^−1^) was reported for *F. antipyretica* from rivers throughout Galicia, Spain [24].

The hypothesis that different moss species can be adopted as biomonitors and can reveal pollution patterns has been already been confirmed [16]. The lack of significant differences in the accumulation of As and Hg between five aquatic bryophytes (*F. antipyretica*, *P. riparioides*, *S. undulata*, *B. rivulare,* and *F. polyphyllus*) justified their combined use as biomonitors [24]. Three moss and two submerged vascular plant species that were applied as biomonitors in the current study confirmed the above-mentioned findings and revealed the possibility of combining aquatic mosses and vascular plants to meet the needs of rapid assessment.

At the same time, species specificity in bioaccumulation should not be neglected. This was evidenced by the established excess of the mercury EQS in *M. spicatum* (0.052 mg kg^−1^) in contrast to the second biomonitor applied at the same site *E. canadensis* (<0.05 mg kg^−1^). Both species are submerged and with a large relative surface area.

### 3.3. Aquatic Macrophytes: Bioindicators and Biomonitors

Macrophyte taxa compositions of the aquatic communities, together with the species’ quantitative relationships, are sensitive to a number of factors, such as flow velocity and shading, diverse pressures, and stressors. The dominance of the referenced species with type-specific abundance reflects the “high” ecological status of the water bodies.

In the study, for the first time, an index based on the indicator role of macrophytes (RI) and an index based on their role as biomonitors (MPI) were applied in parallel. In general, RI quantifies the deviation of aquatic macrophyte communities in terms of species composition and abundance from type-specific reference conditions. MPI, in turn, is more narrowly oriented to evaluate the overall metal contamination. The results indicated a complex relationship between ecological status assessment via RI and MPI. Despite the macrophyte-based index (RI) not being specifically designed to measure toxic impact, it had a good correlation with inorganic pollution revealed by MPI. Nevertheless, the ecological status derived from RI was higher than those derived from MPI, indicating that aquatic macrophytes as biomonitors can add to the precise assessment of stressors affecting environmental conditions in seriously contaminated rivers.

Most of the heavy metals and trace elements have biomagnification ability and reach extremely high bioaccumulated concentrations with increased tropic levels. Priority substances are currently monitored in fish and, as an exception, in invertebrates with the presumption to avoid the risk of secondary poisoning of top predators, including humans. Both aquatic mosses and vascular plants represent a large biological reservoir of heavy metals and trace elements and have an important role in biogeochemical cycling and food chain transfer [21]. In addition, aquatic macrophytes are very suitable for biomonitors in rivers, as they are attached, perennial, and have a high bioaccumulation capacity. Last but not least, their sampling is easy and economical, and mosses are suitable for sampling throughout the year. Thus, studies on aquatic macrophytes as biomonitors should not be neglected as they can serve for establishing EQS for biota at the national level and have the potential to allow for long-term trend analyses of the priority substances, namely, Cd, Pb, Hg, and Ni. Such studies should be replicated in order to verify, expand, and strengthen the achieved results.

## 4. Materials and Methods

The research area was located in the Maritsa River Basin (springing from Rila Mountain) and three of its major tributaries—Chepelarska and Vacha (Rhodopes), and Stryama (Sredna gora Mountains) rivers (Table 6). Shirokolashka River is a right tributary of the Vacha River. All three Rhodopes rivers drain through a zone with deposits of lead–zinc ore. Chepelarska River is additionally under the impact of mine wastes, industrial effluents from the ore-processing factory “Gorubso—Laki”, and the non-ferrous metals plant “KCM—Plovdiv” [18].

Macrophyte surveys were carried out over 100 m, and submerged, free-floating and emergent taxa were considered. Four abiotic parameters were recorded: flow velocity, shading, substrate type, and mean depth [3]. The nomenclature was after Hill et al. [25] for mosses and Euro + Med [26] for vascular plants. Species abundance followed a 5-level scale [27]: 1 = very rare, 2 = infrequent, 3 = common, 4 = frequent, and 5 = abundant and predominant.

Macrophyte-based Reference Index (RI) and EQR were calculated after Gecheva et al. [6]. Element enrichment in plant tissues was based on contamination factor (CF), calculated as the ratio between the element content at a given sampling site and the nearest reference site (3). The adopted by Mouvet [28] scale was applied. Further Metal pollution index (MPI) was calculated [29] and the classification scale after Soares et al. [30] was applied to assess the overall heavy metal and trace element contamination.

Basic physicochemical parameters (e.g., conductivity and pH, Table 1) were measured in situ with a WTW pH/Conductivity meter.

Representative plant samples consisting of 5 to 10 subsamples were collected depending on the stream width and the “patchiness” of the selected biomonitor assemblages [1]. Plants were washed briefly in the stream to remove admixtures. In laboratory conditions, they were cleaned of mineral and organic particles. The air-dried plant samples were prepared for subsequent instrumental analysis by a previously described [31] microwave-assisted acid digestion procedure in closed PTEE vessels. Briefly, 8 mL concentrated HNO_3_ (69% Tracemetal Grade, Thermo Fisher Scientific) were added to the homogenized sample (~0.5 g) and left overnight. Then, 2 mL H_2_O_2_ (pa 35% Sigma Aldrich) was added and samples were subjected to microwave treatment (Ethos 1, Milestone, Denmark). After digestion, samples were transferred and diluted with ultra-pure water (PURELAB Chorus 2+, ELGA Veolia) to the primary dilution factor DF~100. Two types of reference materials, moss RM (M2 and M3) [32,33] and CRM “Bush Branches and Leaves” (NCS DC 73348), were treated according to the same protocol. To control possible contamination due to the reagents or the sample preparation procedure, blank samples were prepared and analyzed for every sample batch as well.

Tested elements were divided into two groups according to the concentration levels in the sample solutions. The concentrations of Al, Fe, Mn at radial, and Zn (at axial plasma) observation mode were determined by ICP-OES (iCAP 6300 Duo, Thermo Scientific Waltham, MA, USA) after additional dilution of the sample by a factor of 20. The determination of trace elements As, Cd, Co, Cr, Cu, Hg, Ni, and Pb was performed by ICP-MS (Agilent 7700, Agilent Technologies, Tokyo, Japan).

To assess the risk of spectral matrix interference in the ICP-OES method, two emission lines were observed for each of the elements, respectively. Al (309.271 and 396.152 nm), Fe (259.94 and 238.204 nm), Mn (259.37 and 259.373 nm), and Zn (213.856 and 202.548 nm) with two isotopes per element (111,113Cd, 52,53Cr, 63,65Cu, 201,202Hg, 60,62Ni, and 206,208Pb) were also monitored in ICP-MS analysis, excluding the monoisotopic elements (75As, 59Co). Additionally, a collision gas (He 4.8 mL/min) was used to eliminate the risk of matrix-induced polyatomic species at As, Cr, Cu, Co, and Ni determinations. To maintain both low methodological limits of trace element quantification and to alleviate the matrix effect, dissolved samples were further diluted by a factor of 5. An internal standard 103Rh was chosen for the online non-spectral matrix effect and instrumental drift correction during ICP-MS measurements.

The standard solutions used for external calibration in both methods were prepared after appropriate dilution of traceable to NIST CRM of 33 components solution (CPAChem, Stara Zagora, Bulgaria) and single-element solutions for Hg and Rh (CPAChem, Stara Zagora, Bulgaria). The results obtained from both analytical methods were validated by the analysis of moss reference materials M2 and M3 and CRM Bush Branches and Leaves.

## Figures and Tables

**Table 1 plants-12-01366-t001:** List of the studied river sites, coordinates (WGS84), altitude, and basic physicochemical parameters. Legend: T—temperature; C—conductivity; DO—dissolved oxygen.

		Coordinates						
	River—Site	N	E	Altitude, m a.s.l.	pH	T, ^°^C	C, µS cm^−1^	DO, mg L^−1^
1	Chepelarska—Kemera	42.1457	24.87722	153	7.9	22	486	7.4
2	Shirokolashka—Breze	41.70222619	24.50111	870	8.3	18.5	222	9.3
3	Vacha—before Teshel Reservoir	41.6679447	24.34574	874	8.2	11.5	192	10.2
4	Shirokolashka—mouth	41.71861531	24.42639	762	8.3	15.5	247	9.3
5	Maritsa—Raduil	42.2758	23.68503	950	6.7	14	33	9.6
6	Stryama—Pesnopoy	42.4747209	24.82083	250	7.6	20	193	8.7

**Table 2 plants-12-01366-t002:** List of the registered aquatic macrophytes and groups with regard to the link to the water after Birk et al. [13]. Legend: BRm—mosses; PHe—helophytes; PHg—hygrophytes; PHy—hydrophytes. A—includes species with high abundance under reference conditions and low abundance or absence under other conditions; B—includes species without preferences for any reference or other conditions; C—encompasses taxa that are rarely found under reference conditions. Usually, they are abundant in habitats where Group A is absent or is poorly represented.

Taxa	Aquaticity Group	Indicator Group (Adapted RI [6])
*Berula erecta* (Huds.) Coville	PHe	B
*Brachythecium rivulare* Schimp.	BRm	A
*Bryum turbinatum* (Hedw.) Turner	BRm	A
*Calamagrostis epigejos* (L.) Roth	PHg	
*Ceratophyllum demersum* L.	PHy	C
*Cinclidotus aquaticus* Bruch & W.P.Schimper	BRm	A
*Elodea canadensis* Michx.	PHy	C
*Epilobium ciliatum* Raf.	PHg	
*Epilobium hirsutum* L.	PHg	
*Epilobium parviflorum* Schreb.	PHg	
*Eupatorium cannabinum* L.	PHe	
*Fontinalis antipyretica* Hedw.	BRm	B
*Leptodictyum riparium* (Hedw.) Warnst.	BRm	B
*Lycopus europaeus* L.	PHe	
*Mentha longifolia* (L.) L.	PHe	
*Myriophyllum spicatum* L.	PHy	C
*Persicaria maculosa* Gray	PHg	
*Phalaris arundinacea* L.	PHe	B
*Plantago lanceolata* L.	PHg	
*Platyhypnidium riparoides* (Hedw.) Dixon	BRm	A
*Potamogeton crispus* L.	PHy	C
*Prunella vulgaris* L.	n.a.	
*Ranunculus aquatilis* L.	PHy	B
*Ranunculus repens* L.	PHg	
*Rorippa sylvestris* (L.) Besser	PHg	
*Rumex* sp.	PHg	
*Saponaria officinalis* L.	PHg	
*Symphytum officinale* L.	PHg	
*Typha latifolia* L.	PHe	
*Urtica dioica* L.	PHg	
*Veronica beccabunga* L.	PHe	

**Table 3 plants-12-01366-t003:** Species richness, abundance, RI, and EQR. Legend: RI—Reference Index; EQR—Ecological Quality Ratio: *blue—high, green—good,* and *yellow—moderate ecological status*.

River Site	Number of Indicator Taxa	Abundance of Indicator Taxa (ABD^3^)	RI	EQR
1	2	16	−50	0.25
2	1	27	0	0.50
3	1	27	100	1.00
4	2	28	96.4	0.98
5	5	33	72.7	0.86
6	5	172	−68.6	0.16

**Table 4 plants-12-01366-t004:** Concentrations (minimum, maximum, and medians) of the analyzed elements in biomonitors, mg kg^−1^.

Element	Min	Species	Max	Species	Median
Al	3627	*F. antipyretica*	17,616	*L. riparium*	4684
As	1.4	*F. antipyretica*	7.2	*L. riparium*	3.6
Cd	0.2	*P. riparioides*, *L. riparium*	64	*L. riparium*	0.6
Co	5.7	*P. riparioides*, *E. canadensis*	14	*L. riparium*	6.3
Cr	3.5	*P. riparioides*, *F. antipyretica*	32	*L. riparium*	7.1
Cu	7.6	*P. riparioides*	114	*L. riparium*	17
Fe	4296	*E. canadensis*	16,242	*L. riparium*	5715
Mn	990	*L. riparium*	6179	*L. riparium*	2200
Ni	3.5	*F. antipyretica*	28	*L. riparium*	7.1
Pb	3.4	*P. riparioides*	916	*L. riparium*	10
Zn	55	*F. antipyretica*	3182	*L. riparium*	102

**Table 5 plants-12-01366-t005:** CFs of 11 heavy metals and trace elements in plant species, and MPI at studied river sites. Legend: green, low contamination; yellow, certain/moderate pollution; orange, strong pollution; red, extreme/heavy pollution.

River Site	Al	As	Cd	Co	Cr	Cu	Fe	Mn	Ni	Pb	Zn	MPI
1	3.8	2.0	262	2.0	9.2	15	2.8	0.8	4.7	270	39	198
2	3.3	1.9	0.8	0.9	5.6	2.6	2.4	0.1	2.8	4.4	1.1	2.1
4	2.0	1.8	1.0	0.8	3.4	5.1	1.4	0.5	1.2	3.2	1.3	1.6
5	0.8	0.4	2.5	0.9	0.8	1.4	0.8	0.2	0.6	2.1	0.7	2.0
6	0.9	0.5	10	0.7	1.8	1.8	0.8	0.3	1.3	3.0	1.3	7.2

**Table 6 plants-12-01366-t006:** Catchment area and length of the studied rivers.

River	Catchment Area, km^2^	Length, km
Vacha	1645	112
Stryama	1395	110
Chepelarska	1010	86
Shirokolashka	218	29

## Data Availability

Not applicable.

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
