# Peer review of "Macrophyte-Based Assessment of Upland Rivers: Bioindicators and Biomonitors"

_plants, 2023, doi:10.3390/plants12061366_

Round 1

Reviewer 1 Report

The manuscript summarizes the results of an aquatic macrophyte sampling campaing developed in some rivers from Bulgary. The topic falls in the scope of Water, it is well-written and properly organized and it is of interest for a broad international audience. Nevertheless some issues have arisen during review and must be corrected before final acceptance (see below).

Lines 54-55: authors include Leptodictyum riparium among the most commonly used species of aquatic mosses for biomonitoring pourpouses. As far as I know, at least if the data gathered and revised by Deben et al. (2015; Ecol. Indic. 53), the use of this species is less common than, as examples, Brachythecium rivulare or even species from genus Scapania. Please, clarify this point and rewrite properly.

Table 1: it would be useful to point out the datum corresponding to these geographical coordinates. I guess it is WGS84 so please include this in table caption.

Line 87: this section (and following ones) shows scientific names in regular fonts nor italized. I suppose this error is due to the document format but revise throughout the ms and check.

Table 3: authors use different colors to highlight EQR values but there is not any explanation about it in caption.

Figure 1: I fully desagree with showing jointly results corresponding to different species. It is known the existence of interspecific differences in, as example, some moss species (F. antipyretica vs. P. riparioides), so to combine their results to draw a single graph it is not advisable. Moreover, in this way readers cannot know the results of each species limiting so the citation of this work in the future. I suggest to remove the figure and show the results in a table, spliting the data of each species (including means, medians, maximum and minimum values). In this way all elements can be showed together and two graphs are not required.

Lines 225-226: as far as I know authors did not use any statistical test for checking differences between species hence is not adequate to say: "no significant diference".

Lines 286 and following: there is not any information about the sampling procedure. Authors state they sampled a 100 m lenght river course but they did not explain if they collected subsamples along this section and then they were combined in a composite sample, or how many subsamples were collected, etc. Please add more details about sampling because the ms. lacks of basic details about sampling strategy.

Lines 309-310: authors used Certified Reference Materials to assess the quality of their analytical procedure but they did not include any explanation about the results (recoveries) obtained. It is essential to include a brief text (maybe a couple of phrases could be enough) regarding those results.

Author Response

We would like to thank the Reviewer 1 for the for the positive evaluation and precise comments. All Reviewer’s recommendations were reflected. (Reviewer’s comments are shown in Italic).

The manuscript summarizes the results of an aquatic macrophyte sampling campaing developed in some rivers from Bulgary. The topic falls in the scope of Water, it is well-written and properly organized and it is of interest for a broad international audience. Nevertheless some issues have arisen during review and must be corrected before final acceptance (see below). Lines 54-55: authors include Leptodictyum riparium among the most commonly used species of aquatic mosses for biomonitoring pourpouses. As far as I know, at least if the data gathered and revised by Deben et al. (2015; Ecol. Indic. 53), the use of this species is less common than, as examples, Brachythecium rivulare or even species from genus Scapania. Please, clarify this point and rewrite properly.

Thanks to your comment the part of the statement related to L. riparium was revised (Line 55-56). Deben et al. (2015) was cited and included in the Reference list.

Table 1: it would be useful to point out the datum corresponding to these geographical coordinates. I guess it is WGS84 so please include this in table caption.

Implemented.

Line 87: this section (and following ones) shows scientific names in regular fonts nor italized. I suppose this error is due to the document format but revise throughout the ms and check.

Thank you - We have checked the manuscript. Latin names are in italics, but probably a change occurred when the file was uploaded. In the pdf file they are saved in italics.

Table 3: authors use different colors to highlight EQR values but there is not any explanation about it in caption.

The color code for the different levels of ecological status was added in the caption.

Figure 1: I fully desagree with showing jointly results corresponding to different species. It is known the existence of interspecific differences in, as example, some moss species (F. antipyretica vs. P. riparioides), so to combine their results to draw a single graph it is not advisable. Moreover, in this way readers cannot know the results of each species limiting so the citation of this work in the future. I suggest to remove the figure and show the results in a table, spliting the data of each species (including means, medians, maximum and minimum values). In this way all elements can be showed together and two graphs are not required.

Following your suggestion, a new table with minimum, maximum and medians and the respective biomonitors (Table 4 in the revised version of the MS) was added to replace Figure 1 (Line 165).

Lines 225-226: as far as I know authors did not use any statistical test for checking differences between species hence is not adequate to say: "no significant diference".

Corrected.

Lines 286 and following: there is not any information about the sampling procedure. Authors state they sampled a 100 m lenght river course but they did not explain if they collected subsamples along this section and then they were combined in a composite sample, or how many subsamples were collected, etc. Please add more details about sampling because the ms. lacks of basic details about sampling strategy.

Thank you – details about sampling were added (please see Line 323-326).

Lines 309-310: authors used Certified Reference Materials to assess the quality of their analytical procedure but they did not include any explanation about the results (recoveries) obtained. It is essential to include a brief text (maybe a couple of phrases could be enough) regarding those results.

Thanks to your recommendation, a paragraph for the obtained percent recoveries (R%) was added (Line 135-143).

Reviewer 2 Report

The work was prepared by the authors very carefully and should be published. However, the literature list lacks works from the last 5 years. Please refer to the latest literature.

Author Response

We thank the Reviewer 2 for the positive evaluation of the manuscript and valuable suggestion. Below you can find the answers to your comments. (Reviewer’s comments are shown in Italic).

The work was prepared by the authors very carefully and should be published. However, the literature list lacks works from the last 5 years. Please refer to the latest literature.

We have searched the databases again, e.g., Scopus, MDPI, but in the last 5 years the researches on aquatic plants have been mainly focused on microplastics and bioremediation – thus we included the review of Demarco et al. (2023).

Reviewer 3 Report

Based on my understanding of the manuscript, the goal of this study was to use ecological and environmental assessment measurements and evaluations to assess upland rivers.  

Overall, I thought the general manuscript idea was sound and interesting to a broad readership and somewhat novel.  However, I thought the execution of writting of the paper and presentation of the results could be improved.

I found the authors use of pronouns such as they and their when refering to plants to be unclear and urge the authors to avoid pronouns to be more clear.

In the introduction, I thought the authors would help the readers by providing some background and detail on the various evaluation metrics, even thought those were discussed more in the methods.   Part of the issue for me, and this may be this journals style, is that the methods were presented after the results and discussion, so when I got to the results, I had a hard time following the results without a frame of reference. 

For the results.  Table 1, would it be helpful to provide drainage areas or stream orders part of the description to give the reader a sense of size/scale.  Also, I felt the results were vague and did not include abunances or relative abundances of the various taxa to help evaluate the EQR calculation prior to the presentation of the EQR table 3.  For the chemical assessment, I thought there was not enough data presented that reflected the text, so I really didnt follow the authors findings.  Further, Figure 1 and table 4 didnt seem to make sense or I could not follow the calculation steps going into table 4.

Discussion. While I thought the discussion was one of the stronger sections of the paper, the authors mentioned a hypothesis, but I didnt notice an explicit hypothesis in the introduction,  and really, in the introduction, I would have like to see clear question, goal, objectives and stated hypothesis.

Methods and Materials.  Not sure if this is the style of this journal, but the methods were after the discussion, so much of the detail about the various indices and assessments were found much later in the paper after the presentation of the results, and those details would have helped me prepare for critically evaluating and understanding the presented findings.

Author Response

We thank the Reviewer 3 for the valuable comments. Below you can find the answers to your comments. (Reviewer’s comments are shown in Italic).

Based on my understanding of the manuscript, the goal of this study was to use ecological and environmental assessment measurements and evaluations to assess upland rivers. Overall, I thought the general manuscript idea was sound and interesting to a broad readership and somewhat novel. However, I thought the execution of writting of the paper and presentation of the results could be improved. I found the authors use of pronouns such as they and their when refering to plants to be unclear and urge the authors to avoid pronouns to be more clear.

Thank you – the usage of “they” and ‘their” arose out of a desire not to repeat words. Thanks to your comment, they have been replaced where possible.

In the introduction, I thought the authors would help the readers by providing some background and detail on the various evaluation metrics, even thought those were discussed more in the methods.   Part of the issue for me, and this may be this journals style, is that the methods were presented after the results and discussion, so when I got to the results, I had a hard time following the results without a frame of reference. 

Thank you – we followed the Journal’s template and section “Materials and methods” has to be the last one (before References).

For the results.  Table 1, would it be helpful to provide drainage areas or stream orders part of the description to give the reader a sense of size/scale.  Also, I felt the results were vague and did not include abunances or relative abundances of the various taxa to help evaluate the EQR calculation prior to the presentation of the EQR table 3. 

Following your recommendation, a new table containing rivers’ catchment areas and length was added (Table 6 in the revised version of the MS). The abundance of indicator taxa was included in Table 3.

For the chemical assessment, I thought there was not enough data presented that reflected the text, so I really didnt follow the authors findings.  Further, Figure 1 and table 4 didnt seem to make sense or I could not follow the calculation steps going into table 4.

Thank you. Figure 1 was replaced with a new table with minimum, maximum and median values of the analyzed elements - Table 4 in the revised version of the MS. Calculation of CFs and MPI followed Mouvet (1986), Gonçalves et al. (1992) Soares et al. (1999) – as described in Materials and Methods.

Discussion. While I thought the discussion was one of the stronger sections of the paper, the authors mentioned a hypothesis, but I didnt notice an explicit hypothesis in the introduction,  and really, in the introduction, I would have like to see clear question, goal, objectives and stated hypothesis.

Thank you – as we stated in the Introduction: “The purpose was to illustrate the complementary nature of aquatic macrophyte application both as a part of ecological status metrics and as a matrix for priority substances monitoring. If our hypothesis is correct, and if upland rivers (where bryophytes are the dominant flora) are under toxic stressors, then accumulation in biota should be added to the final assessment of the ecological status. The hypothesis implicit is that, in the case of chemical pollution to which mosses are tolerant, the macrophyte-based index should be supported with accumulation in biota data.”

Methods and Materials.  Not sure if this is the style of this journal, but the methods were after the discussion, so much of the detail about the various indices and assessments were found much later in the paper after the presentation of the results, and those details would have helped me prepare for critically evaluating and understanding the presented findings.

We followed the Journal’s template and the section “Materials and methods” has to be the last one (before References). This section was enriched in the revised version with details.

Round 2

Reviewer 3 Report

It appears the authors addressed my concerns.